# Allelopathic interactions of *Carthamus oxyacantha, Macrophomina phaseolina* and maize: Implications for the use of *Carthamus oxyacantha* as a natural disease management strategy in maize

Nazir Aslam[1], Muhammad Akbar[1]*, Anna Andolfi[2,3]

**1** Department of Botany, University of Gujrat, Gujrat, Punjab, Pakistan, **2** Department of Chemical Sciences, University of Naples Federico II, Complesso Universitario di Monte Sant'Angelo Via Cintia, Napoli, Italy, **3** BAT Center-Interuniversity Center for Studies on Bioinspired Agro-Environmental Technology, University of Naples Federico II, Portici, Italy

* muhammad.akbar@uog.edu.pk

**Data Availability Statement:** All relevant data are within the manuscript and its Supporting information files.

## Abstract

Fungicides are used to control phytopathogens but all these fungicides have deleterious effects. Allelopathic interactions can be harnessed as a natural way to control the pathogens but there are no reports that show the allelopathic interactions of donor plant, recipient crop, as well as the target plant pathogen and the material used for inoculum production. So, in the present study, the suitability of *Carthamus oxyacantha* M. Bieb. was assessed against *Macrophomina phaseolina*, the cause of charcoal rot in maize. Among the various treatments in pot experiment, a negative control, 3 concentrations of inoculum ($1.2×10^5$, $2.4×10^5$, and $3.6×10^5$ colony forming units (CFU) mL$^{-1}$, 3 concentrations (0.5, 1.0, and 1.5% w/w) of *C. oxyacantha* along with an autoclaved *M. phaseolina* (Mp) and *C. oxyacantha* alone were included to investigate their allelopathic effects on maize, not investigated earlier. Maximum suppression of the disease was observed by 1.5% (w/w) concentration of *C. oxyacantha*. Soil amendment with *C. oxyacantha* significantly suppressed the disease incidence (DI) and disease severity index (DSI) in charcoal rot of maize up to 40 and 55%, respectively over the strongest level of inoculum (Mp3). *C. oxyacantha* not only reduced area under disease incidence progress curve (AUDIPC) and area under disease severity progress curve (AUDSPC), but also improved the morphological, biochemical and physiological parameters of maize. The maximum increase of 48, 65, and 75% in values of shoot length (SL), shoot dry mass (SDM), and root dry mass (RDM), respectively was observed by application of the highest concentration of *C. oxyacantha* in the treatment Mp1+Co3, over infested control (Mp1). Photosynthetic pigments, such as chlorophyll *a*, chlorophyll *b* and carotenoids were increased to 58, 64, and 46%, respectively over Mp1, by the application of *C. oxyacantha*. Carbon assimilation rate (*A*), stomatal conductance (*gs*), rate of transpiration (*E*), and internal carbon dioxide concentration (*Ci*) were significantly increased to 58, 48, 48, and 20%, respectively over infested control (Mp3), by application of *C. oxyacantha* concentration 1.5 (w/w). Moreover, defense enzymes like superoxide dismutase

**Funding:** The author(s) received no specific funding for this work.

**Competing interests:** The authors have declared that no competing interests exist.

(SOD), peroxidase (POD) and catalase (CAT) activities were boosted up to 27, 28, and 28% over Mp3, respectively. Positive allelopathy of *C. oxyacantha* towards maize and negative allelopathy towards *M. phaseolina* makes *C. oxyacantha* a suitable candidate for charcoal rot disease control in maize.

## Introduction

Maize (*Zea mays* L.) is a staple food crop which stands at 1st position with respect to its production, while it stands at 3rd position regarding its cultivation worldwide, after wheat and rice [1]. Pakistan stands on the 18th position among the maize producing countries, with 5.27 t ha$^{-1}$ yield, that is very low as compared to per hectare yield, 9.5 t ha$^{-1}$ in United States [2, 3]. There are many factors involved in this low per hectare yield, including non-availability of high yielding maize varieties, costly fertilizers, poor implementation of agricultural policies, adverse climatic conditions, and biotic constraints [4].

Pathogenic fungi are the main biotic constraints causing plant diseases and resulting in 31% yield loss in maize [5]. Among them, *Macrophomina phaseolina* (Tassi) Gold, infects over 500 host plants with morphological symptoms such as, charcoal rot, canker, damping off and blights [6].

Charcoal rot of maize has developed into more alarming situation because of resistance found in pathogen against environmental factors, ability to cause epidemic and its propensity to kill the host plants, causing up to 63.6% yield loss in the maize crop [7, 8]. *M. phaseolina* is a soil and seed borne pathogen, its transmission from seed to seedling has also been reported [9]. Pakistan is one amongst the countries which are facing devastating environmental changes. Rise in average daily temperature of Pakistan, up to 0.87 ºC is slightly more than average global increase. Meanwhile, Pakistan experiences some of the highest maximum temperatures in the world, with average maximum temperature of 38 ˚C and above in many regions [10]. So, these climate changes are very suitable for growth of *M. phaseolina*, the causal agent of charcoal rot disease in crops.

Various strategies such as avoidance, breeding for disease resistance, cultural practices and chemical disease management can be adopted to control charcoal rot of maize. However, the presence of natural antifungal compounds in plants is an emerging technology to manage plant diseases [11–14]. Several successful investigations reported that application of plant extracts, residue, compost and mulches used either *in vitro* or *in vivo* significantly controlled various fungal pathogens [15–18]. Previous *in vitro* studies revealed that *C. oxyacantha* has antifungal activity against various fungal pathogens such as *Aspergillus niger*, *M. phaseolina*, *Rhizoctonia solani*, *Fusarium oxysporum*, and mushrooms [19].

*C. oxyacantha* (wild safflower, Family: Asteraceae), is an annual herbaceous weed plant with white stem and simple sessile spiny leaves. *C. oxyacantha* is abundantly found in wheat fields in the plains of Punjab, Pakistan, with orange yellow capitulum and achene fruit. *C. oxyacantha* contains antifungal compounds e.g., D-Ribofuranose, 5-deoxy-5-(methylsulfinyl)-1,2,3-tris-O-(trimethylsilyl), Benzoic acid, 4-hydroxy-3-methoxy-, methyl ester, and γ-Sitosterol [20, 21]. Along with presence of antifungal compounds in an organic material to cope with fungal diseases in plants, its compatibility with crop plants is also very important. Allelopathy of plant residue must be checked before its application in field [22]. Although, *in vitro* antifungal activity of *C. oxyacantha* has been reported, it's *in vivo* effectiveness against charcoal rot of maize as well as its compatibility for maize is missing. Therefore, a pot experiment was

conducted to investigate the compatibility and antifungal efficacy of *C. oxyacantha* to control charcoal rot of maize caused by *M. phaseolina*. Moreover, effect of soil amendment with *C. oxyacantha* on the morphological, physiological, and defense related attributes of maize were also investigated for the first time in this study.

## Materials and methods

### Collection of indigenous weed, pathogen and test plant

Indigenous weed, *C. oxyacantha* was collected from District Mandi Bahauddin, Punjab Province, Pakistan, and was identified on the basis of vegetative and floral characters, and compared with literature [23]. *C. oxyacantha* is an herbaceous plant having spiny-leaves alternately arranged on non-woody stem, having deep lobe with dentate margins. Orange colored flowers grow as flower head, having size of 2 to 3 cm in diameter. The plants were air-dried under shade. Dried plants were ground to powder with the help of mini electric grinder Nm-8300, and stored at room temperature in polythene bags.

Certified seeds of maize, variety Neelam, were purchased from the local market. The isolate of fungal pathogen, *Macrophomina phaseolina* was isolated from diseased maize plant collected from maize field, village Sagera, District Kasur, Punjab, Pakistan. For the Identification of fungal isolate, the color of culture, diameter of colony (cm), and the size of microsclerotia were noted [24]. *M. phaseolina* was identified as gray color colony, maximum colony size was 80 to 85 mm after 7 days of incubation, whereas the size of microsclerotia was 48.00 $\mu$m in diameter.

Sorghum seeds were used as a substrate for multiplication of fungal inoculum. Sorghum seeds were soaked overnight in 4 L of a solution containing distilled water and 40 g of sucrose, 0.5 g of yeast extract and 0.25 g of tartaric acid per liter. The solution was decanted, and the sorghum seeds were divided equally into autoclavable bags. A plastic tube of 5 cm in diameter and 10 cm long was inserted halfway into the bags for the placement of culture plugs onto the sorghum seeds. Cotton plugs were inserted into each tube, and the samples were autoclaved at 121 ˚C for 30 min. One-week-old culture plugs of *M. phaseolina*, grown on PDA (Potato dextrose agar), were used to inoculate the sorghum seeds. Mycelial plugs were placed into each bag, and each opening was then reclosed with the cotton plug. The bags were incubated at 28 ±2 ˚C for 3 weeks, with periodic shaking to spread the inoculum on sorghum seeds within the bags. After 3 weeks, the sorghum seeds were completely colonized and darkened with fungal hyphae/microsclerotia. The dried sorghum based inoculum was stored in sealed plastic containers at 4 ˚C until further use [25].

### *In vivo* assessment of antifungal activity of *C. oxyacantha*

Antifungal efficacy of *C. oxyacantha* was investigated in a pot experiment (Pots were kept in open air to simulate natural field conditions). Whereas, pots were used to enhance inoculum equally for equable infection and better control of *M. phaseolina* with *C. oxyacantha* in a completely randomized design (CRD), having 19 treatments with 5 replicates. Corn field soil and pots (20 cm in diameter and 30 cm deep) were sterilized by 5% formalin solution for 15 minutes and left to dry for two weeks. Soil amendments with selected weed, *C. oxyacantha* were made with three concentrations (0.5, 1.0 and 1.5% w/w), by mixing the *C. oxyacantha* in pots of selected treatments. Amended soil was added in each pot (7 kgs/pot). Sorghum based fungal inoculum [26] was ground to form hyphal/microsclerotial suspension in sterilized distilled water and three concentrations *viz*., $1.2{\times}10^5$ CFU mL$^{-1}$, $2.4{\times}10^5$ CFU mL$^{-1}$ and $3.6{\times}10^5$ CFU mL$^{-1}$, respectively were maintained by serial dilution method. Pot soil of selected treatments was infested by inoculum of the pathogen by mixing ten mL of hyphal/microsclerotia

**Table 1. Treatments composition in pot experiment.**

| Sr. No. | Treatments | Description |
|---|---|---|
| 1 | C | Control (Without pathogen and soil amendment) |
| 2 | Mp1 | *Macrophomina phaseolina* ($1.2 \times 10^5$ CFU mL$^{-1}$) |
| 3 | Mp2 | *M. phaseolina* ($2.4 \times 10^5$ CFU mL$^{-1}$) |
| 4 | Mp3 | *M. phaseolina* ($3.6 \times 10^5$ CFU mL$^{-1}$) |
| 5 | AMp1 | Autoclaved *M. phaseolina* ($1.2 \times 10^5$ CFU mL$^{-1}$) |
| 6 | AMp2 | Autoclaved *M. phaseolina* ($2.4 \times 10^5$ CFU mL$^{-1}$) |
| 7 | AMp3 | Autoclaved *M. phaseolina* ($3.6 \times 10^5$ CFU mL$^{-1}$) |
| 8 | Co1 | *Carthamus oxyacantha* 0.5% (w/w) |
| 9 | Co2 | *C. oxyacantha* 1% (w/w) |
| 10 | Co3 | *C. oxyacantha* 1.5% (w/w) |
| 11 | Mp1+Co1 | *M. phaseolina* ($1.2 \times 10^5$ CFU mL$^{-1}$)+*C. oxyacantha* 0.5% |
| 12 | Mp1+Co2 | *M. phaseolina* ($1.2 \times 10^5$ CFU mL$^{-1}$)+*C. oxyacantha* 1% |
| 13 | Mp1+Co3 | *M. phaseolina* ($1.2 \times 10^5$ CFU mL$^{-1}$)+*C. oxyacantha* 1.5% |
| 14 | Mp2+Co1 | *M. phaseolina* ($2.4 \times 10^5$ CFU mL$^{-1}$)+*C. oxyacantha* 0.5% |
| 15 | Mp2+Co2 | *M. phaseolina* ($2.4 \times 10^5$ CFU mL$^{-1}$)+*C. oxyacantha* 1% |
| 16 | Mp2+Co3 | *M. phaseolina* ($2.4 \times 10^5$ CFU mL$^{-1}$)+*C. oxyacantha* 1.5% |
| 17 | Mp3+Co1 | *M. phaseolina* ($3.6 \times 10^5$ CFU mL$^{-1}$)+*C. oxyacantha* 0.5% |
| 18 | Mp3+Co2 | *M. phaseolina* ($3.6 \times 10^5$ CFU mL$^{-1}$)+*C. oxyacantha* 1% |
| 19 | Mp3+Co3 | *M. phaseolina* ($3.6 \times 10^5$ CFU mL$^{-1}$)+*C. oxyacantha* 1.5% |

suspension in topsoil of 10 cm depth. Healthy maize seeds were sown on 5[th] day of inoculation [27], at the rate of 3 seeds/pot and each pot was thinned to one seedling at ten days after emergence. Standard agronomical conditions were maintained [28]. In total 19 treatments were made. The detailed composition of treatments in pot experiment is given in Table 1.

## Disease assessment

DI and DSI were assessed simultaneously, three times during the whole pot experiments, with the interval of 14 days. Disease symptoms for charcoal rot of maize, appearance of lesion at the collar region [27] were observed 42 days after sowing (DAS), at growth stage 3, (Collar of 12[th] leaf visible, leaves 3 and 4 may be dead) in plants of infested pots. Two more successive observations were made with the interval of 14 days, at growth stage 4, 56 DAS (Collar of 14[th] leaves visible, tips of many tassels visible) and at growth stage between 5 and 6, 70 DAS (75% of plants have silks visible) [29]. Both DI and DSI were evaluated at the same time. Depending on the disease symptoms, DSI was scaled on a 0–5 scale [30]. Numerical disease rating was assigned as follows: 0, healthy plants; 1, appearance of lesion at the collar region, 2–7 mm in length; 2, large lesions, 8–12 mm in length; 3, moderate rotting of the collar region, loss of turgor at the top with slight drooping; 4, extensive rotting at the collar region, wilting and drying of many leaves, drooping of the shoot; 5, plants completely wilted, dead and dry. DI and DSI were measured by following equations;

$$\text{DI} = \frac{\text{No. of diseased plants}}{\text{Total number of Plants}} \times 100 \tag{1}$$

$$\text{DSI} = \frac{0 \times \text{P0} + 1 \times \text{P1} + 2 \times \text{P2} + 3 \times \text{P3} + 4 \times \text{P4} + 5 \times \text{P5}}{\text{N(G} - 1)} \times 100 \tag{2}$$

Where P0 to P5 are total number of observed plants in each disease grading per treatment, N is total number of observations and G stands for number of grading. The AUDIPC and AUDSPC both were calculated by [31].

$$AUDIPC/AUDSPC = \sum_{n=1}^{n-1}(X_i + Xi - 1/2)(Ti - Ti - 1) \tag{3}$$

Where Xi is $1^{st}$ reading of disease incidence/disease severity and Xi+1 indicated each successive reading of disease incidence/disease severity at time (t).

## Morphological parameters

Morphological parameters were measured after harvesting the plants on 80 DAS. SL, SDM, and RDM of all plants in all treatments were recorded after oven drying at 70 ˚C until a constant dry weight reading was achieved [32].

## Estimation of photosynthetic pigments

The measurement of photosynthetic pigments such as chlorophyll *a*, *b*, and carotenoids was done at 42 DAS. For this, 100 mg of apical leaves from each treatment (parts of three leaves were randomly mixed) were cut into small pieces and mixed with 5 mL of 80% acetone in triplicates. Homogenization was done in a pre-cooled sterile mortar and pestle. The obtained extract was then centrifuged at 3000 rpm for 15 min and the clear solution was transferred to a new vial with a final volume made up to 5 mL by 80% acetone. For these three pigments, the optical density was recorded at 663, 645, and 440.5 nm wavelength by using a spectrophotometer (Model UV 3000), respectively [33]. The levels of chlorophyll *a*, *b* and carotenoids were measured by the following equations;

$$Chlorophyll\ a\ mg\ g^{-1} = 12.7\ (OD)\ 663 - 2.69(OD)645 \times (w/v \times 1000) \tag{4}$$

$$Chlorophyll\ b\ mg\ g^{-1} = 22.9\ (OD)\ 645 - 4.68\ (OD)\ 663 \times (w/v \times 1000) \tag{5}$$

$$Total\ carotenoids\ mg\ g^{-1} = 46.95\ (OD)\ 440.5\ - (0.268 \times chl\ a\ +\ b) \tag{6}$$

where v = final volume (mL) of extract in 80% acetone, w = fresh weight of leaf in grams.

## Physiological measurements

Physiological measurements were made after 56 DAS on the $6^{th}$ leaf of plants from each treatment (5 replications per treatment) with help of a portable infrared gas analyzer (IRGA) (model: ADC-USA 1264) and the following four parameters were calculated: Net carbon assimilation rate (A) $\mu$mol $CO_2$ $m^{-2}s^{-1}$; Stomatal conductance (*gs*) mmol $m^{-2}s^{-1}$; Internal $CO_2$ concentration (*Ci*); Transpiration rate (*E*) mmol $H_2O$ $m^{-2}s^{-1}$ [37]. The measurements were made between 10:00 AM and 2:00 PM.

## Measurements of antioxidant activities

For SOD, POD and CAT antioxidant activities, 0.5 g of fresh leaf tissues (mixed from 10 leaves collected from plants for each treatment at 56 DAS were ground into a fine powder by pre chilled mortar and pestle. The leaf powder was homogenized by adding 3 mL of chilled 100 mM PBS buffer (pH 7.8) [34]. After adding 1.5 mL of homogenate in the two centrifuge tubes, the supernatant was centrifuged at 10,000 *x g* for 20 min at 4 ˚C. Centrifuged supernatant was transferred to new centrifuge tubes for further analysis.

**SOD activity assay.** Solution mixture (for 95 reactions) were prepared by adding 95 mL 100 mM PBS (pH 7.8), 1.9 mL 1 mM EDTA, 6.4 mL 130 mM Met, 6.4 mL 750 μM NBT, and 6.4 mL 20 μM Riboflavin. Crude enzyme solution (50 μL) from each sample was added into 1 mL reaction solution in a 1.5 mL centrifuge tube. Reaction solution with 50 μL 100 mM PBS (pH 7.8) but no crude enzyme under dark and light condition served as controls I and control II, respectively. All the tubes were exposed to the light intensity of 4,000 lux for 10–15 min, except the control I which was kept in the dark, while other tubes were quickly moved away from the light. Spectrophotometer absorbance was measured at 560 nm in the dark and used control I as reference [35].

$$SOD\ total\ activity\ (unit:\ u/gFW) = [(AC - As)\ x\ V]\ /\ (0.5\ x\ Ack\ x\ Vt)/FW \qquad (7)$$

Ac: Control II, absorbance at 560 nm, $A_S$: sample tube, absorbance at 560 nm, V: total volume of enzyme solution, Vt: volume of enzyme used in the test tube, FW: fresh weight of sample (g).

**POD activity assay.** To determine POD activity, solution mixture (for 95 reactions) was prepared by adding 53.2 μL 0.2% guaiacol in 95 mL 100 mM PBS (pH 7.0), heated and stirred well, then added 36.1 μL 30% $H_2O_2$ after cooling. 50 μL 100 mM PBS (pH 7.8) and 1 mL of the reaction solution were mixed into a cuvette for reference (control) [35, 36].

$$POD\ activity\ (unit:\ u/gFW) = \Delta A470\ x\ (V/Vt)/(0.01\ x\ t)/FW \qquad (8)$$

$\Delta A470$: the change in absorbance at 470 nm during every 20 seconds, V: total volume of enzyme solution, Vt: volume of enzyme used in cuvette, t: time of reaction (min), FW: sample fresh weight (g).

**CAT activity assay.** Solution mixture (for 95 reactions) was prepared by adding 147.25 μL 30% $H_2O_2$ in 95 mL 100 mM Phosphate buffer solution (PBS, pH 7.0). 50 μL crude enzyme and 1 mL of the reaction solution were taken in the cuvette and the absorbance at 240 nm was recorded immediately with spectrophotometer at every 15 seconds for 1 min, by looking for steady average alteration. Reaction solution with 50 μL 100 mM PBS (pH 7.8) was used as a reference.

$$CAT\ activity\ (unit:\ u/mg\ protein) = \Delta A240\ x\ (V/Vt)/(0.1\ x\ t)/FW \qquad (9)$$

$\Delta A240$: the change of absorbance at 240 nm during every 15 seconds [35, 36]. V: total volume of crude enzyme solution, Vt: volume of crude enzyme used in the test tube t: reaction time (min), FW: fresh weight (g).

## Statistical analysis

For statistical analysis, ANOVA was done followed by Fisher's LSD test at 5% probability using computer software Minitab 20. Principal component analysis (PCA) biplot was performed by using OriginPro 2024.

## Results

### Disease assessment

Data regarding disease assessment are given in Table 2. DI and DSI were measured at three different stages, 42 DAS, 56 DAS, and 70 DAS. There were no disease symptoms in negative control and treatments with only *C. oxyacantha*. When comparing three different strengths of the inoculum, DI and DSI recorded for Mp1, Mp2 and Mp3 increased progressively with increasing inoculum levels from 60 up to 100% for DI and from 16 up to 88% for DSI, respectively.

**Table 2. Effect of different treatments on disease incidence (DI), disease severity index (DSI), area under disease incidence progress curve (AUDIPC), and area under disease severity progress curve (AUDSPC) on maize plants, grown in pots.**

| Treatments | Disease Incidence (%) | | | AUDIPC (%) | Disease Severity Index (%) | | | AUDSPC (%) |
|---|---|---|---|---|---|---|---|---|
| | 42DAS | 56DAS | 70DAS | | 42DAS | 56DAS | 70DAS | |
| Mp1 | 60±24.49 a-c | 60±24.49 a-c | 80±20.00 ab | 1820±610.0 a-c | 16±7.5 b-d | 24±11.66 b-e | 44±14.70 c-e | 756±312 b-d |
| Mp2 | 80±20.00 ab | 80±20 ab | 80±20 ab | 2240±560 ab | 24±7.4 b | 48±13.56 ab | 80±20.00 ab | 1400±362 ab |
| Mp3 | 100±0.00 a | 100±0.00 a | 100±0.00 a | 2800±0.0 a | 56±11.7 a | 68±12.0 a | 88±4.90 a | 1960±221 a |
| Mp1+Co1 | 40±24.49 b-d | 60±24.49 a-c | 60±24.49 a-c | 1540±560.0 a-c | 12±8.0 b-d | 24±11.7 b-e | 32±13.56 c-f | 644±305 c-e |
| Mp1+Co2 | 20±20.00 cd | 40±24.49 b-d | 40±24.49 b-d | 980±610 b-d | 8±8.0 b-d | 16±9.80 c-e | 24±14.60 d-f | 448±278 de |
| Mp1+Co3 | 20±20.00 cd | 20±20.00 cd | 20±20.00 cd | 560±560.0 cd | 4±4.0 cd | 12±12.00 de | 16±16.00 ef | 308±308 de |
| Mp2+Co1 | 60±24.49 a-c | 60±24.49 a-c | 80±20.49 ab | 1820±610 a-c | 20±10.95 bc | 28±12.00 b-e | 52±13.6 b-d | 896±315 b-d |
| Mp2+Co2 | 40±24.49 b-d | 40±24.49 b-d | 60±.24.49 a-c | 1260±641.6 b-d | 12±8.0 b-d | 24±14.7 b-e | 36±16.0 c-e | 672±358 c-e |
| Mp2+Co3 | 20±20.00 cd | 20±20.00 cd | 40±24.49 b-d | 700±542.3 cd | 4±4.0 cd | 12±12.0 de | 28±17.44 d-f | 392±298 de |
| Mp3+Co1 | 80±20.00 ab | 80±20 ab | 80±20.00 ab | 2240±560.0 ab | 24±7.5 b | 44±11.66 a-c | 64±16.00 a-c | 1232±317 bc |
| Mp3+Co2 | 60±24.49 a-c | 60±24.49 a-c | 80±20.00 ab | 1820±610.3 a-c | 16±7.5 b-d | 32±13.56 b-d | 44±11.66 c-e | 868±308 b-d |
| Mp3+Co3 | 40±24.49 b-d | 40±24.49 b-d | 60±24.49 a-c | 1260±641.6 b-d | 12±8.0 b-d | 28±17.44 b-e | 40±16.73 c-e | 756±388 b-d |

**Mp1;** *Macrophomina phaseolina* ($1.2\times10^5$ CFU mL$^{-1}$), **Mp2;** *Macrophomina phaseolina* ($2.4\times10^5$ CFU mL$^{-1}$), **Mp3;** *Macrophomina phaseolina* ($3.6\times10^5$ CFU mL$^{-1}$), **Co1;** *Carthamus oxyacantha* 0.5%, **Co2;** *Carthamus oxyacantha* 1%, **Co3;** *Carthamus oxyacantha* 1.5%, **DAS**; days after sowing, **AUDIPC**; Area under disease incidence progress curve, **AUDSPC**; Area under disease severity progress curve. Data presented represent means ± standard error of 5 replicates, followed by a different alphabet, differ significantly at $P < 0.05$ according to Fisher's LSD test.

**Note:** Disease symptoms of charcoal rot did not appear in treatments *viz.*, **C;** control, **AMp1;** Autoclaved *Macrophomina phaseolina* ($1.2\times10^5$ CFU mL$^{-1}$), **AMp2;** Autoclaved *Macrophomina phaseolina* ($2.4\times10^5$ CFU mL$^{-1}$), **AMp3;** Autoclaved *Macrophomina phaseolina* ($3.6\times10^5$ CFU mL$^{-1}$), **Co1, Co2,** and **Co3.** So, these data are excluded from the table.

Soil amendments with Co1, Co2, and Co3 dry *C. oxyacantha* significantly reduced the DI and DSI over positive controls with all tested strengths of inoculum. This antifungal efficacy of *C. oxyacantha* was reduced with the passage of time but increased with increasing quantity of the *C. oxyacantha* in respective treatments. Finally, at 70 DAS, among three tested concentrations of *C. oxyacantha*, (Mp3+Co3) suppressed DI and DSI in charcoal rot of maize up to 40 and 55%, respectively over the strongest level of inoculum (Mp3). AUDIPC and AUDSPC were calculated from three consecutive readings with 14 days of interval. Both AUDIPC and AUDSPC for Mp1, Mp2, and Mp3 were increased from 1820 up to 2800% for AUDIPC and from 756 up to 1960% for AUDSPC, respectively. Finally, at 70 DAS, among three tested concentrations of *C. oxyacantha*, (Mp3+Co3) significantly suppressed AUDIPC and AUDSPC in charcoal rot of maize up to 55 and 61%, respectively, over the strongest level of the inoculum, Mp3 (Table 2, S1 and S2 Files).

## Effects of treatments with *Carthamus oxyacantha* on morphological attributes of maize

Data regarding the effects of treatments on the morphological attributes of *C. oxyacantha* are presented in (Fig 1A–1C). SL, SDM, and RDM were significantly decreased by the application of three different inoculum levels (Mp1, Mp2, and Mp3), up to (25, 29, and 32%), (26, 39, and 48%), and (30, 44, and 52%), respectively, over control (C). Whereas, soil amendment with *C. oxyacantha* with three concentrations (Co1, Co2, and Co3) increased SL, SDM, and RDM up to (4, 12, and 15%), (8, 21, and 26%), and (13, 17, and 30%), respectively over C. Application of *C. oxyacantha* in infected treatments also significantly increased SL, SDM, and RDM. The minimum increase in the values of above mentioned parameters was 29, 56, and 64%,

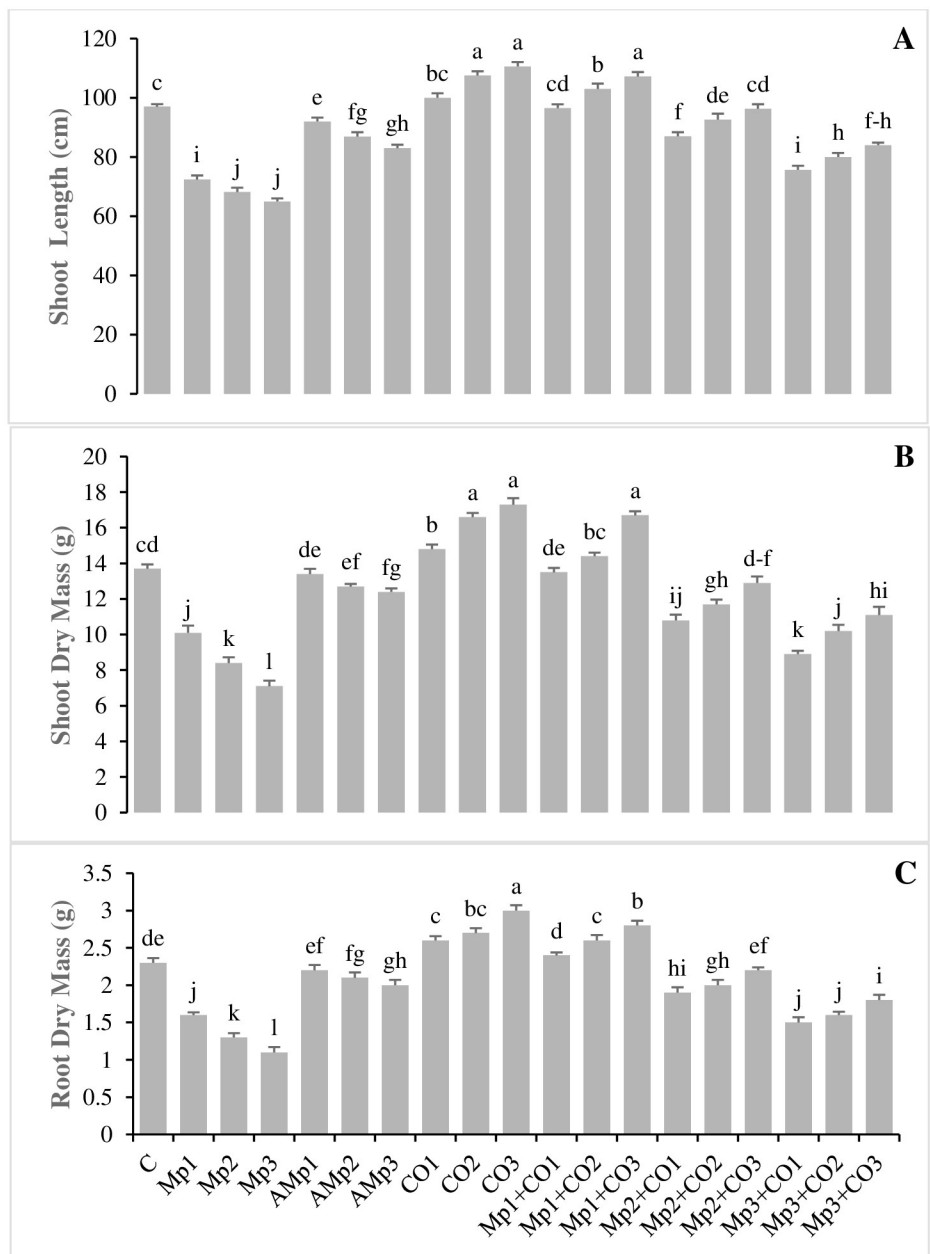

**Fig 1.** Effect of treatments on (**A**) shoot length, (**B**) shoot dry mass, and (**C**) root dry mass of maize in pot trials. Data represent means ± standard error of 5 replicates. Error bars with a common alphabet do not differ significantly at *P* = 5% as computed by Fisher's LSD test, using Minitab 20.2. *Abbreviations*: **C**: control (Without pathogen and soil amendment), **Mp**: *Macrophomina phaseolina*, **AMp**: Autoclaved *M. phaseolina*, **Co**: *Carthamus oxyacantha*, **Mp1**: Mp (1.2×10$^5$), **Mp2**: Mp (2.4×10$^5$), **Mp3**: Mp (3.6×10$^5$), **AMp1**: AMp (1.2×10$^5$), **AMp2**: AMp (2.4×10$^5$), **AMp3**: AMp (3.6×10$^5$), **Co1**: Co0.5%, **Co2**: Co1%, **Co3**: Co1.5%, **Mp1+Co1**: Mp (1.2×10$^5$)+Co0.5%, **Mp1+Co2**: Mp (1.2×10$^5$) +Co1%, **Mp1+Co3**: Mp (1.2×10$^5$)+Co1.5%, **Mp2+Co1**: Mp (2.4×10$^5$)+Co0.5%, **Mp2+Co2**: Mp (2.4×10$^5$)+Co1%, **Mp2 +Co3**: Mp (2.4×10$^5$)+Co1.5%, **Mp3+Co1**: Mp (3.6×10$^5$)+Co0.5%, **Mp3+Co2**: Mp (3.6×10$^5$)+Co1%, **Mp3+Co3**: Mp (3.6×10$^5$)+ Co1.5%. Note: Mp concentrations are given in colony forming units (CFU mL$^{-1}$).

respectively by application of the highest concentration of the treatment Mp3+Co3, over the highest level of inoculum Mp3. While, maximum increase of 48, 65, and 7% in the values of aforesaid parameters, respectively was observed by application of the highest concentration of *C. oxyacantha* in the treatment Mp1+Co3, over Mp1 (Fig 1A–1C, S3 File).

## Effects of treatments with *Carthamus oxyacantha* on pigments of maize

Data about the effects of soil amendments on the pigments of *C. oxyacantha* are shown in (Fig 2A–2C). Photosynthetic pigments, such as chlorophyll *a* (Chl *a*), chlorophyll *b* (Chl *b*) and carotenoids were decreased by the application of three levels of inoculum (Mp1, Mp2, and Mp3). This decrease was up to (31, 37, and 28%), (38, 49, and 35%), and (41, 55, and 39%) in Chl *a*, Chl *b*, and carotenoids, respectively over C. However, the values of Chl *a*, Chl *b*, and carotenoids were significantly increased in the treatments having *C. oxyacantha* amendments over the infected treatments (Mp1, Mp2 and Mp3). The minimum increase in the values of Chl *a*, Chl *b*, and carotenoids was seen in the treatment Mp3+Co3, which were 48, 58, and 34% for Chl *a*, Chl *b*, and carotenoids, respectively, over Mp3. Whereas, the maximum increase in the values of Chl *a*, Chl *b*, and carotenoids was seen in the treatment Mp1+Co3, which was 58, 64, and 46% for Chl *a*, Chl *b*, and carotenoids, respectively, over the positive control treatment (Mp1) (Fig 2A–2C, S4 File).

## Effects of treatments with *Carthamus oxyacantha* on physiological attributes of maize

Data regarding the effects of different treatments on the physiological attributes of *C. oxyacantha* are presented in (Figs 3A, 3B, 4A & 4B). (*A*) and (*gs*) were decreased up to (28, 34, and 47%) and (29, 33, and 45%), respectively for inoculated treatments (Mp1, Mp2, and Mp3), in comparison with non-inoculated treatment C. A significant decrease in (*A*) and (*gs*), were also observed for the treatments with autoclaved inoculums (Amp2 and Amp3). There was an increase of (2, 10, and 16%) for (*A*) with the treatments Co1, Co2, and Co3 whereas, values of (*gs*) were significantly increased for Co2 and Co3 up to 17 and 25%, respectively, for treatments with *C. oxyacantha* (Co2, and Co3), in comparison to treatment C. There was maximum increase of 32, 45, and 54%, respectively for (*A*); 28, 44, and 58%, respectively for (*gs*), by treatments with *C. oxyacantha* (Mp1+Co1, Mp1+Co2, and Mp1+Co3), in comparison to inoculated treatment Mp1. There was minimum but significant increase of 22, 34, and 40%, respectively for (*A*); 22, 38, and 48%, respectively for (*gs*), by treatments with *C. oxyacantha* (Mp3 +Co1, Mp3+Co2, and Mp3+Co3) in comparison with inoculated treatment Mp3 (Fig 3A & 3B, S5 File).

A significant decrease of 20, 26, and 38% in (*E*) was observed with the application of three different levels of inoculum such as Mp1, Mp2, and Mp3, respectively. Whereas, with the application of three different levels of *C. oxyacantha* Co1, Co2, and Co3, the values of (*E*) progressively increased up to 5, 13, and 28% over C, respectively. Minimum increase of up to 3, 14, and 43% in infested treatments (Mp3+Co1, Mp3+Co2, and Mp3+Co3) was observed over inoculated treatment Mp3. Whereas, the maximum increase of up to 9, 25, and 48% in infested treatments (Mp1+Co1, Mp1+Co2, and Mp1+Co3) was observed over inoculated treatment Mp1 (Fig 4A).

Carbon dioxide concentration (*Ci*) significantly increased up to 14, 25, and 32% with the application of Mp1, Mp2, and Mp3, respectively over C. Application of three different levels of *C. oxyacantha* Co1, Co2, and Co3 also increased (*Ci*) up to 11, 17, and 21%, respectively. Minimum significant increase of up to 7, 10, and 14% in infested treatments (Mp3+Co1, Mp3 +Co2, and Mp3+Co3) was observed over inoculated treatment, Mp3. Whereas, the maximum

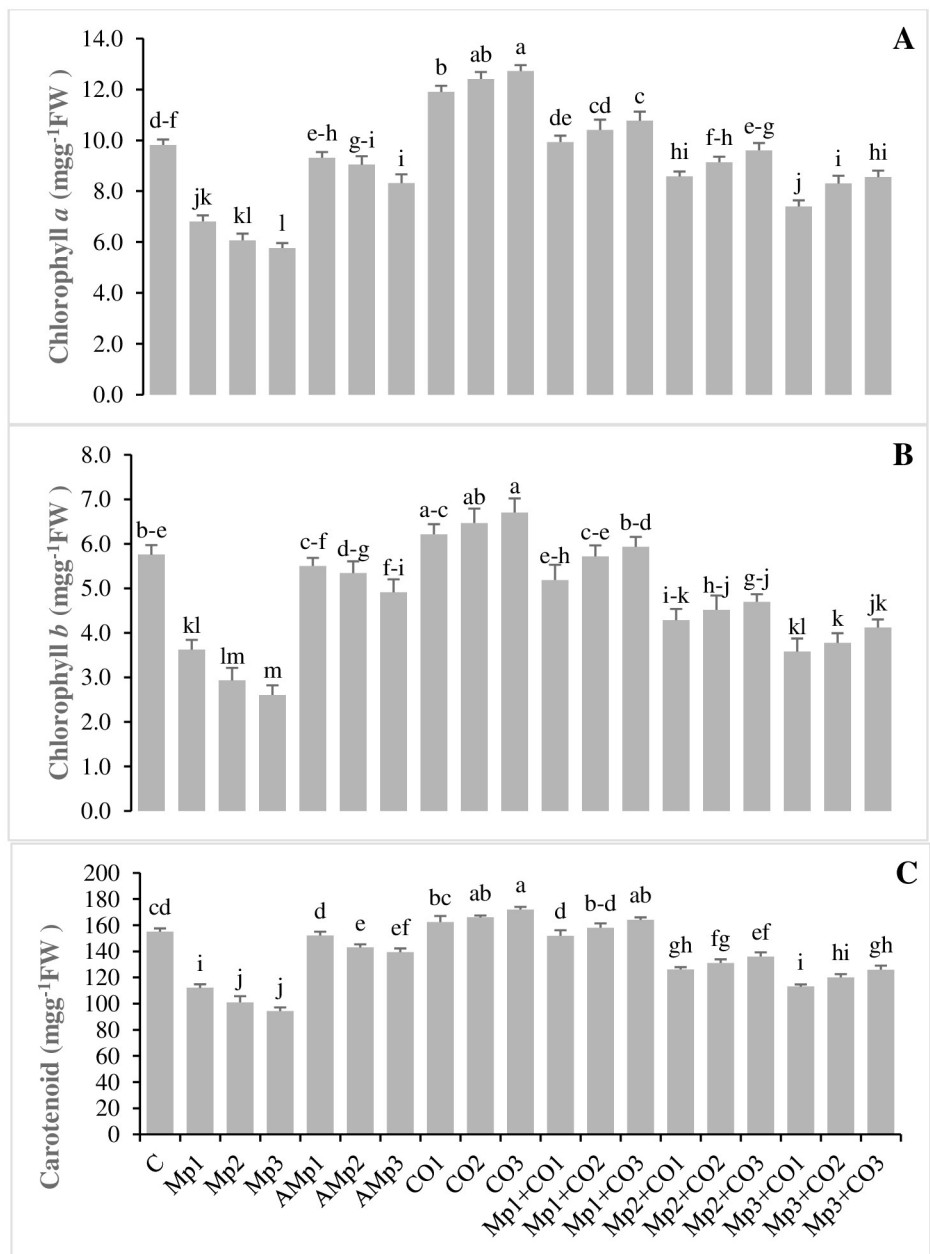

**Fig 2.** Effect of treatments on **(A)** chlorophyll *a*, **(B)** chlorophyll *b*, and **(C)** carotenoids of maize in pot trials. Data represent means ± standard error of 5 replicates. Error bars with a common alphabet do not differ significantly at *P* = 5% as computed by Fisher's LSD test, using Minitab 20.2. *Abbreviations*: **C**: control (Without pathogen and soil amendment), **Mp**: *Macrophomina phaseolina*, **AMp**: Autoclaved *M. phaseolina*, **Co**: *Carthamus oxyacantha*, **Mp1**: Mp ($1.2 \times 10^5$), **Mp2**: Mp ($2.4 \times 10^5$), **Mp3**: Mp ($3.6 \times 10^5$), **AMp1**: AMp ($1.2 \times 10^5$), **AMp2**: AMp ($2.4 \times 10^5$), **AMp3**: AMp ($3.6 \times 10^5$), **Co1**: Co0.5%, **Co2**: Co1%, **Co3**: Co1.5%, **Mp1+Co1**: Mp ($1.2 \times 10^5$)+Co0.5%, **Mp1+Co2**: Mp ($1.2 \times 10^5$) +Co1%, **Mp1+Co3**: Mp ($1.2 \times 10^5$)+Co1.5%, **Mp2+Co1**: Mp ($2.4 \times 10^5$)+Co0.5%, **Mp2+Co2**: Mp ($2.4 \times 10^5$)+Co1%, **Mp2 +Co3**: Mp ($2.4 \times 10^5$)+Co1.5%, **Mp3+Co1**: Mp ($3.6 \times 10^5$)+Co0.5%, **Mp3+Co2**: Mp ($3.6 \times 10^5$)+Co1%, **Mp3+Co3**: Mp ($3.6 \times 10^5$)+ Co1.5%. Note: Mp concentrations are given in colony forming units (CFU mL$^{-1}$).

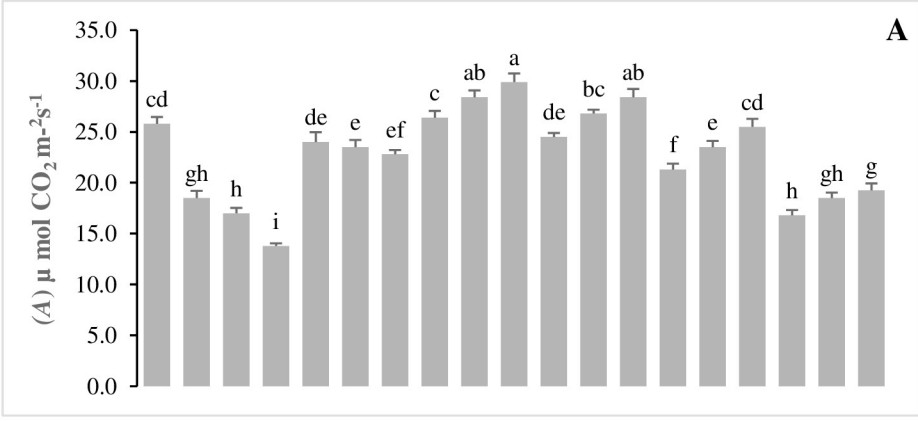

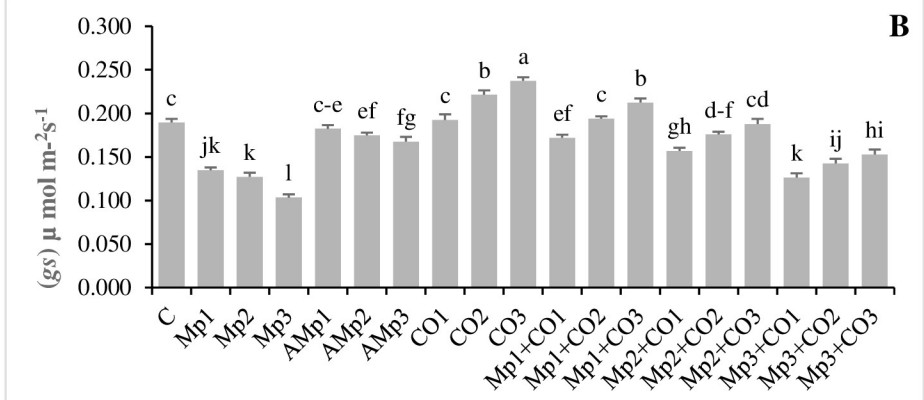

**Fig 3.** Effect of treatments on **(A)** rate of carbon assimilation, and **(B)** stomatal conductance of maize in pot trials. Data represent means ± standard error of 5 replicates. Error bars with a common alphabet do not differ significantly at $P = 5\%$ as computed by Fisher's LSD test, using Minitab 20.2. *Abbreviations*: **C**: control (Without pathogen and soil amendment), **Mp**: *Macrophomina phaseolina*, **AMp**: Autoclaved *M. phaseolina*, **Co**: *Carthamus oxyacantha*, **Mp1**: Mp $(1.2 \times 10^5)$, **Mp2**: Mp $(2.4 \times 10^5)$, **Mp3**: Mp $(3.6 \times 10^5)$, **AMp1**: AMp $(1.2 \times 10^5)$, **AMp2**: AMp $(2.4 \times 10^5)$, **AMp3**: AMp $(3.6 \times 10^5)$, **Co1**: Co0.5%, **Co2**: Co1%, **Co3**: Co1.5%, **Mp1+Co1**: Mp $(1.2 \times 10^5)$+Co0.5%, **Mp1+Co2**: Mp $(1.2 \times 10^5)$ +Co1%, **Mp1+Co3**: Mp $(1.2 \times 10^5)$+Co1.5%, **Mp2+Co1**: Mp $(2.4 \times 10^5)$+Co0.5%, **Mp2+Co2**: Mp $(2.4 \times 10^5)$+Co1%, **Mp2 +Co3**: Mp $(2.4 \times 10^5)$+Co1.5%, **Mp3+Co1**: Mp $(3.6 \times 10^5)$+Co0.5%, **Mp3+Co2**: Mp $(3.6 \times 10^5)$+Co1%, **Mp3+Co3**: Mp $(3.6 \times 10^5)$+ Co1.5%. Note: Mp concentrations are given in colony forming units (CFU mL$^{-1}$).

increase of up to 15, 19, and 20% in infested treatments (Mp1+Co1, Mp1+Co2, and Mp1 +Co3) was observed over Mp1. There was significant decrease of 5 and 10%, for ($Ci$), for treatments (Amp2 and Amp3), respectively, over C (Fig 4B, S6 File).

## Effects of treatments with *Carthamus oxyacantha* on defense related antioxidant enzymes of maize

Data regarding the effects of different treatments on the activities of antioxidant enzymes are given in (Fig 5A–5C). Activities of antioxidant enzymes were noticeably enhanced in infected maize plants. SOD activity was significantly increased up to 32, 55, and 79% due to pathogen infection for Mp1, Mp2, and Mp3, respectively over C. Application of *C. oxyacantha* also significantly increased the values of SOD, up to 14, 25, and 39%, over C respectively, for Co1, Co2, and Co3. The minimum increase of 9% in the SOD activity was observed in the treatment

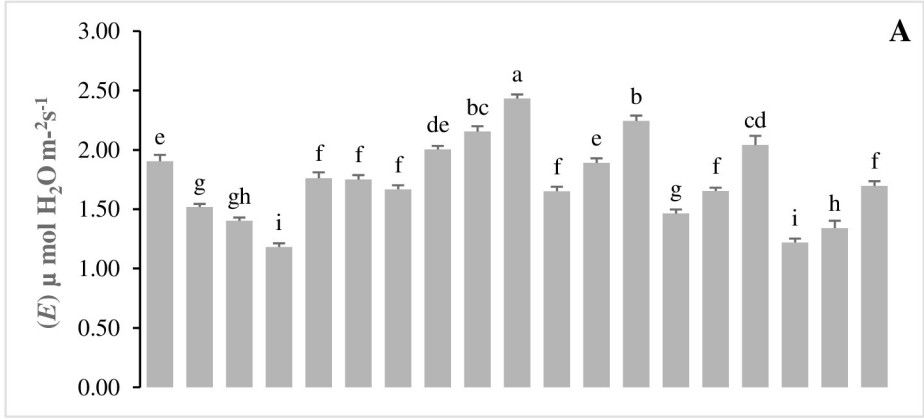

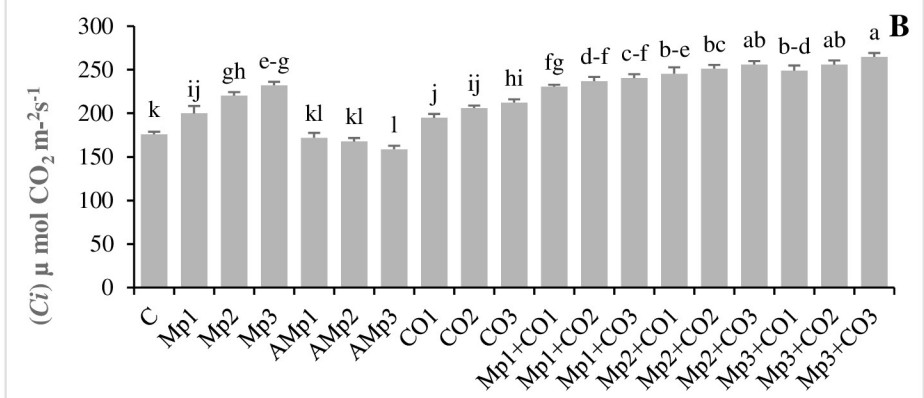

**Fig 4.** Effect of treatments on **(A)** transpiration rate, and **(B)** internal carbon dioxide conc. of maize in pot trials. Data represent means ± standard error of 5 replicates. Error bars with a common alphabet do not differ significantly at $P$ = 5% as computed by Fisher's LSD test, using Minitab 20.2. *Abbreviations*: **C**: control (Without pathogen and soil amendment), **Mp**: *Macrophomina phaseolina*, **AMp**: Autoclaved *M. phaseolina*, **Co**: *Carthamus oxyacantha*, **Mp1**: Mp $(1.2 \times 10^5)$, **Mp2**: Mp $(2.4 \times 10^5)$, **Mp3**: Mp $(3.6 \times 10^5)$, **AMp1**: AMp $(1.2 \times 10^5)$, **AMp2**: AMp $(2.4 \times 10^5)$, **AMp3**: AMp $(3.6 \times 10^5)$, **Co1**: Co0.5%, **Co2**: Co1%, **Co3**: Co1.5%, **Mp1+Co1**: Mp $(1.2 \times 10^5)$+Co0.5%, **Mp1+Co2**: Mp $(1.2 \times 10^5)$ +Co1%, **Mp1+Co3**: Mp $(1.2 \times 10^5)$+Co1.5%, **Mp2+Co1**: Mp $(2.4 \times 10^5)$+Co0.5%, **Mp2+Co2**: Mp $(2.4 \times 10^5)$+Co1%, **Mp2 +Co3**: Mp $(2.4 \times 10^5)$+Co1.5%, **Mp3+Co1**: Mp $(3.6 \times 10^5)$+Co0.5%, **Mp3+Co2**: Mp $(3.6 \times 10^5)$+Co1%, **Mp3+Co3**: Mp $(3.6 \times 10^5)$+ Co1.5%. Note: Mp concentrations are given in colony forming units (CFU mL$^{-1}$).

Mp1+Co1, over Mp1. The maximum increase in the value of SOD was 26.7% for Mp3+Co3, over Mp3 (Fig 5A).

POD activity was also significantly enhanced due to pathogen infection up to 9, 17, and 25% for Mp1, Mp2, and Mp3, respectively over C. The application of *C. oxyacantha* also increased the values of POD significantly, up to 5 and 13% over C respectively, for Co2 and Co3. The minimum increase (17%) in the value of POD activity was seen in the treatment Mp1+Co1, over Mp1, while, the maximum increase was 28% for Mp3+Co3, over Mp3 (Fig 5B).

CAT activities were also increased in infected treatments up to 47, 67, and 95% for Mp1, Mp2, and Mp3, respectively over C. Application of *C. oxyacantha* also increased the values of CAT significantly, up to 36, 46, and 63% over C, respectively, for Co1, Co2, and Co3. The minimum increase (25.2%) in CAT activity was seen in the treatment Mp1+Co1, over Mp1. On the contrary, the maximum increase was 28% for Mp3+Co3, over Mp3 (Fig 5C, S7 File).

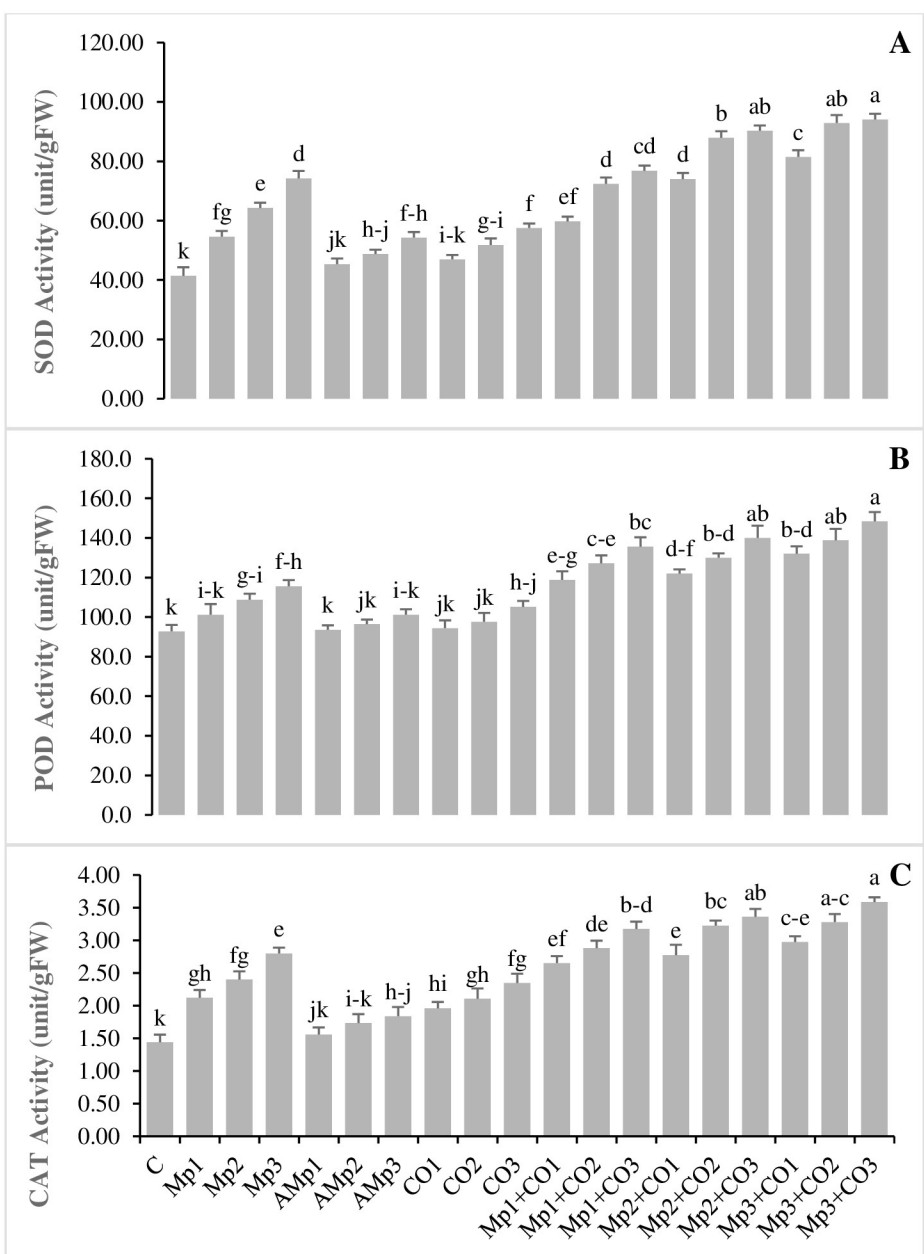

**Fig 5.** Effect of treatments on **(A)** superoxide dismutase, **(B)** peroxidase, and **(C)** catalase activities of maize in pot trials. Data represent means ± standard error of 5 replicates. Error bars with a common alphabet do not differ significantly at *P* = 5% as computed by Fisher's LSD test, using Minitab 20.2. *Abbreviations*: **C**: control (Without pathogen and soil amendment), **Mp**: *Macrophomina phaseolina*, **AMp**: Autoclaved *M. phaseolina*, **Co**: *Carthamus oxyacantha*, **Mp1**: Mp ($1.2 \times 10^5$), **Mp2**: Mp ($2.4 \times 10^5$), **Mp3**: Mp ($3.6 \times 10^5$), **AMp1**: AMp ($1.2 \times 10^5$), **AMp2**: AMp ($2.4 \times 10^5$), **AMp3**: AMp ($3.6 \times 10^5$), **Co1**: Co0.5%, **Co2**: Co1%, **Co3**: Co1.5%, **Mp1+Co1**: Mp ($1.2 \times 10^5$)+Co0.5%, **Mp1 +Co2**: Mp ($1.2 \times 10^5$)+Co1%, **Mp1+Co3**: Mp ($1.2 \times 10^5$)+Co1.5%, **Mp2+Co1**: Mp ($2.4 \times 10^5$)+Co0.5%, **Mp2+Co2**: Mp ($2.4 \times 10^5$)+Co1%, **Mp2+Co3**: Mp ($2.4 \times 10^5$)+Co1.5%, **Mp3+Co1**: Mp ($3.6 \times 10^5$)+Co0.5%, **Mp3+Co2**: Mp ($3.6 \times 10^5$) +Co1%, **Mp3+Co3**: Mp ($3.6 \times 10^5$)+ Co1.5%. Note: Mp concentrations are given in colony forming units (CFU mL$^{-1}$).

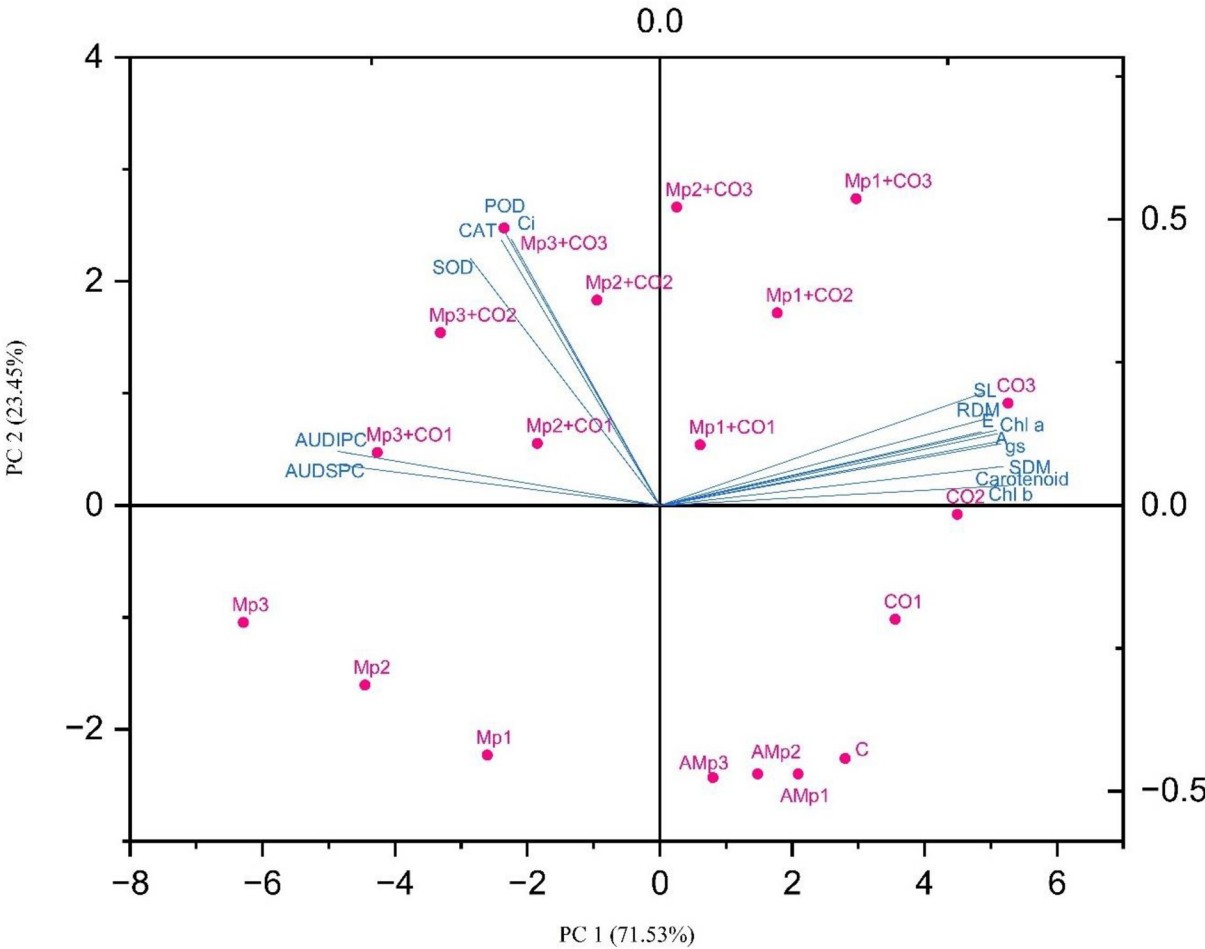

**Fig 6. Principal component analysis (PCA) biplot performed by using OriginPro 2024.** Abbreviations of treatments: C: control (Without pathogen and soil amendment), Mp: *Macrophomina phaseolina*, AMp: Autoclaved *M. phaseolina*, Co: *Carthamus oxyacantha*, Mp1: Mp $(1.2\times10^5)$, Mp2: Mp $(2.4\times10^5)$, Mp3: Mp $(3.6\times10^5)$, AMp1: AMp $(1.2\times10^5)$, AMp2: AMp $(2.4\times10^5)$, AMp3: AMp $(3.6\times10^5)$, Co1: Co0.5%, Co2: Co1%, Co3: Co1.5%, Mp1+Co1: Mp $(1.2\times10^5)$+Co0.5%, Mp1+Co2: Mp $(1.2\times10^5)$+Co1%, Mp1+Co3: Mp $(1.2\times10^5)$+Co1.5%, Mp2+Co1: Mp $(2.4\times10^5)$+Co0.5%, Mp2+Co2: Mp $(2.4\times10^5)$+Co1%, Mp2+Co3: Mp $(2.4\times10^5)$+Co1.5%, Mp3+Co1: Mp $(3.6\times10^5)$+Co0.5%, Mp3+Co2: Mp $(3.6\times10^5)$+Co1%, Mp3+Co3: Mp $(3.6\times10^5)$+ Co1.5%. Note: Mp concentrations are given in colony forming units (CFU mL$^{-1}$). Abbreviations of parameters: SL: Shoot length, SDM: Shoot dry mass, RDM: Root dry mass, Chl *a*: Chlorophyll *a*, Chl *b*: Chlorophyll *b*, *A*: rate of carbon assimilation, *gs*: stomatal conductance, *E*: transpiration rate, *Ci*: Internal carbon dioxide concentration, SOD: superoxide dismutase, POD: peroxidase, CAT: catalase, AUDIPC: Area under disease incidence progress curve, AUDSPC: Area under disease severity progress curve.

Principal component analysis (PCA) biplot showing the relationships of treatments and effects is shown in Fig 6 (S1 Table).

## Discussion

In the present study, soil amendment with an astraceous weed *C. oxyacantha* was investigated against *M. phaseolina*, the cause of charcoal rot of maize variety (Neelam). The effect of different treatments was assessed in terms of morphological, physiological, and biochemical attributes of maize plants under pot conditions, to investigate efficacy of antifungal weed against charcoal rot of maize. Disease assessment of charcoal rot infection in maize variety Neelam was based on the variables of DI, DSI, AUDIPC, and AUDSPC. The occurrence of first symptom of charcoal rot was observed on infected plants near the tasseling stage of maize. Both DI

and DSI were measured at three different times with 14 days of interval. AUDIPC and AUDSPC were calculated on 70 DAS. There was no disease in negative control maize plants. When comparing three different strengths of inoculum, DI, DSI, AUDIPC, and AUDSPC recorded for Mp1, Mp2, and Mp3 were increased with increase of inoculum level of the pathogen in the soil. The remarkable infection level in infected plants may be linked to interruption of water and mineral upward flow in xylem vessels, owing to colonization of the stalk tissues by *M. phaseolina*. Previous reports demonstrated that the flow of minerals and water was damaged by the colonization of *Fusarium verticillioides* in the conducting tissues of maize plants [37]. The similar interruption by infection of *Ceratocystis fimbriata* and *Ceratocystis smalleyi* in mango and bitternut hickory plants was reported by earlier workers [37, 38]. Previously, the disease severity of olive leaf spot (OLS) was increased when inoculum concentration of *Spilocaea oleaginea* increased from $1.0{\times}10^2$ to $2.5{\times}10^5$ conidia $mL^{-1}$ [39]. In the present pot assays, antifungal activity of *C. oxyacantha* with concentrations 0.5, 1, and 1.5% (W/W) was assessed against the charcoal rot infection of maize caused by *M. phaseolina*. When different concentrations of *C. oxyacantha* as soil amendments were used to control the disease infection, percentage of DI and DSI were decreased with increasing concentrations of *C. oxyacantha*. Similar findings were reported in a previous study where root rot incidence of cowpea was decreased by the application of neem leaves [40]. It was also observed in our study that percentages of DI and DSI were increased with increasing days after sowing (DAS) in infected treatments. *C. oxyacantha* having antifungal potency against charcoal rot, suppressed DI and DSI up to 40 and 55%, over Mp3, respectively, in treatment (Mp3+Co3). AUDIPC and AUDSPC both were decreased with increasing concentrations of *C. oxyacantha* in selected treatments. In another study, soil amendment with neem cake and farmyard manure reduced the incidence of charcoal rot disease in chickpea and soybean [41].

Soil amendments with organic materials in crop fields is an efficient way to restore soil organic matter content and to improve soil quality directly or indirectly. Directly it is due to enhancing the availability of micro and macronutrients and indirectly by providing some biologically active compounds (antifungal compounds and antioxidants). Soil amendments with organic materials may alter the communities of microorganisms, which might be helpful to suppress harmful effects of other pathogens. Previous investigations indicate that organic amendments can reduce the incidence of diseases caused by soil borne pathogens, including *M. phaseolina* [11, 42]. Soil amendments with different plant residues have different effects on the growth of other plants either by enhancing the growth of recipient plant or by retarding the growth of recipient plant. In present investigation, *C. oxyacantha* showed positive allelopathy on maize plants as well as antifungal efficacy to control charcoal rot in maize. Application of *C. oxyacantha* increased SL, SDM, and RDM, over non-infested control (C). In a previous investigation [43], reported the positive allelopathy of *C. oxyacantha* on maize plants. Similarly, the application of dry leaf powder of *Acacia nilotica* L. at the rate of 1, 2 and 3% (W/W) in mash bean infested by *M. phaseolina*, increased SL, SFW, SDW, RFW, and RDW up to 35, 96, 45, 92, and 74%, over non-treated infested control [44].

In general, the autoclaved *M. phaseolina* also negatively affected the morphological, biochemical, and physiological attributes of maize and this effect was significant at the higher concentrations. It might be due to sorghum seeds used as a substrate for inoculum multiplication as negative allelopathy of sorghum was reported in a previous study [45]. Moreover, the negatively allelopathy of sorghum for maize has already reported earlier [46]. The photosynthetic pigments such as chlorophyll *a*, chlorophyll *b*, and carotenoids were significantly reduced in infected plants over healthy plants (control) in pots. Previous investigations also reported that fungal infection had inhibitory effects on the synthesis of chlorophyll as well as carotenoids [47]. Previously, it has been demonstrated that infection of *M. phaseolina* in mung bean

reduced the content of photosynthetic pigments [12]. This reduction in synthesis of photosynthetic pigments is due to maximum utilization of energies to combat against fungal infection rather in the synthesis of pigments by the infected plants. In the present study, soil amendments with *C. oxyacantha*, enhanced the content of photosynthetic pigments in maize plants. Previously, increase in photosynthetic pigments by the addition of organic matter in soil was reported [14, 48].

Physiochemical alterations may also occur in plants by pathogens causing root infection and disturb the vascular system which ultimately affect plant growth by hindering the rate of photosynthesis and carbon assimilation by the plants [37, 39]. For many host-pathogen interactions, a decrease in (*A*) was linked with lower (*gs*) and higher (*Ci*) [37, 49]. The photosynthetic restrictions can also be attributed to limitations in $CO_2$ fixation at the biochemical level, not only to reductions in $CO_2$ influx due to stomatal closure [37, 50]. Interestingly, in the present study, there was a progressive decline in (*A*), (*gs*), and (*E*) values while, the (*Ci*) values increased in maize leaves in response to infection. Stomatal limitations may have contributed to the lower (*A*) values since it was accompanied by decrease in (*gs*) values indicating, therefore, an imperceptible influx of $CO_2$ into the leaf tissues that could impair photosynthesis in leaves as a result of *M. phaseolina* infection in the stalk tissues. In the present study, maize plants infected with *M. phaseolina* resulted in great dehydration promoted by fungal colonization of the xylem vessels affecting (*gs*), (*Ci*), and (*E*), consequently lowering (*A*) due to stomatal closure [37, 51]. An interesting aspect *M. phaseolina* infection was an increase in (*Ci*) values indicating a behavior associated with lower activity of photosynthetic enzymes (e.g., RuBisCo) limiting $CO_2$ fixation at the chloroplasts level in wheat plants by the infection of *Pyricularia oryzae* [52]. Moreover, reductions in (*A*) due to *M. phaseolina* infection are mainly associated with stomatal limitations coupled to loss of biochemical performance in the photosynthetic process. Application of soybean cake also increased in (*A*), (*gs*), and (*E*), by 21, 22, and 21%, respectively [53]. Similarly, soil amendment with moringa leaves in maize field increased (*Ci*) to 62%, over control [54].

Soil amendments with *C. oxyacantha*, enhanced the SOD, POD, and CAT activities. The harmful effects of reactive oxygen species (ROS) are minimized by increased activities of antioxidant enzymes such as SOD, POD, and CAT. In fact, SOD, POD, and CAT activities were increased in the maize plant tissues by *M. phaseolina*. In the present study, the addition of pathogen inoculum as well as amendment with *C. oxyacantha* resulted in higher SOD, POD, and CAT activities. The higher antioxidant activities are considered beneficial for the plant as these enzyme activities help to remove the ROS generated in infected plant tissues or by soil amendments with plant residues. Similarly, previous investigations reported that SOD and POD activities were increased in maize plants infected by *F. oxysporum* [37]. Previously it has been demonstrated that SOD activities were increased in wheat plants by the infection of *M. phaseolina* [50]. Similarly, charcoal rot infection increased POD activities in infected plants up to 15%, over non-infested control [11]. Present results of increase in CAT activities by pathogen infection are in agreement with previous studies [11, 55]. Growth enhancement effects of *C. oxyacantha* on maize and growth retarding effects on *M. phaseolina* makes *C. oxyacantha* a suitable soil amendment for charcoal rot disease control in maize. Moreover, the negative allelopathy depicted by the autoclaved inoculum on carrier material suggests to evaluate the effect of these amendments in soil amendment assays to decipher the false positive or negative results.

## Conclusion

In present investigation, *C. oxyacantha* showed strong antifungal activity against *M. phaseolina*, causing charcoal rot in maize. Antifungal efficacy of *C. oxyacantha* was increased by

increasing its concentration. Addition of *C. oxyacantha* not only suppressed the charcoal rot in maize, it also enhanced SL, SDM and RDM of maize plants. Photosynthetic pigments were also increased in infested and non-infested plants by the addition of *C. oxyacantha*. Moreover, physiological parameters, (*A*), (*gs*), (*E*), and (*Ci*) were also enhanced in infested plants by soil amendment with *C. oxyacantha*. Activities of SOD, POD, and CAT were also increased in infested and non-infested plants by the addition of *C. oxyacantha*. Disease suppressing ability of *C. oxyacantha* suggests that soil amendment with *C. oxyacantha* can be used against *M. phaseolina*. Additionally, positive allelopathy of *C. oxyacantha* on maize plants indicate that it might have nutrients or growth stimulating substances which enhanced growth parameters in non-infested maize plants. The treatments included in the present investigation to evaluate the individual effects of Amp as well as *C. oxyacantha* strongly suggest to include the appropriate controls of treatments in order to avoid false positive or negative results in soil amendment bioassays.

## Recommendations

As soil amended with *C. oxyacantha* effectively controlled charcoal rot of maize in pot experiment and also showed compatibility with maize plants, therefore, *C. oxyacantha* can be utilized by farmers to control charcoal rot and increase per hectare yield in their fields.

## Supporting information

**S1 File. ANOVA file for the effect of treatments on the disease incidence (DI) and area under disease progress curve disease incidence (AUDPC DI) on maize plants.**
(DOCX)

**S2 File. ANOVA file for the effect of treatments on the disease severity (DS) and area under disease progress curve disease severity (AUDPC DI) on maize plants.**
(DOCX)

**S3 File. ANOVA file for the effect of treatments on the morphological attributes of maize.**
(DOCX)

**S4 File. ANOVA file for the effect of treatments on the pigments of maize.**
(DOCX)

**S5 File. ANOVA file for the effect of treatments on (A) rate of carbon assimilation and (B) stomatal conductance of maize.**
(DOCX)

**S6 File. ANOVA file for the effect of treatments on (A) transpiration rate and (B) internal carbon dioxide concentration of maize.**
(DOCX)

**S7 File. ANOVA file for the effect of treatments on (A) superoxide dismutase, (B) peroxidase, and (C) catalase activities of maize.**
(DOCX)

**S1 Table. Values used for Principal Component Analysis (PCA).**
(DOCX)

## Author Contributions

**Conceptualization:** Muhammad Akbar.

**Formal analysis:** Nazir Aslam.

**Methodology:** Nazir Aslam.

**Resources:** Nazir Aslam.

**Supervision:** Muhammad Akbar, Anna Andolfi.

**Validation:** Anna Andolfi.

**Writing – original draft:** Nazir Aslam.

**Writing – review & editing:** Muhammad Akbar.

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
