## [Decision Letter · Decision Letter 0]

9 Jan 2024

PONE-D-23-34629Carthamus oxyacantha as bio-elicitor and natural defense in maize against charcoal rot disease caused by Macrophomina phaseolinaPLOS ONE

Dear Dr. Akbar,

Thank you for submitting your manuscript to PLOS ONE. After careful consideration, we feel that it has merit but does not fully meet PLOS ONE’s publication criteria as it currently stands. Therefore, we invite you to submit a revised version of the manuscript that addresses the points raised during the review process. Please submit your revised manuscript by Feb 23 2024 11:59PM. If you will need more time than this to complete your revisions, please reply to this message or contact the journal office at plosone@plos.org. Please include the following items when submitting your revised manuscript:A rebuttal letter that responds to each point raised by the academic editor and reviewer(s). You should upload this letter as a separate file labeled 'Response to Reviewers'.A marked-up copy of your manuscript that highlights changes made to the original version. You should upload this as a separate file labeled 'Revised Manuscript with Track Changes'.An unmarked version of your revised paper without tracked changes. You should upload this as a separate file labeled 'Manuscript'.

We look forward to receiving your revised manuscript.

Kind regards,

Abhay K. Pandey

Academic Editor

PLOS ONE

Journal Requirements:

2. Thank you for submitting the above manuscript to PLOS ONE. During our internal evaluation of the manuscript, we found significant text overlap between your submission and previous work in the [introduction, conclusion, etc.].

Please revise the manuscript to rephrase the duplicated text, cite your sources, and provide details as to how the current manuscript advances on previous work. Please note that further consideration is dependent on the submission of a manuscript that addresses these concerns about the overlap in text with published work.

[If the overlap is with the authors’ own works: Moreover, upon submission, authors must confirm that the manuscript, or any related manuscript, is not currently under consideration or accepted elsewhere. If related work has been submitted to PLOS ONE or elsewhere, authors must include a copy with the submitted article. Reviewers will be asked to comment on the overlap between related submissions (http://journals.plos.org/plosone/s/submission-guidelines#loc-related-manuscripts).]

We will carefully review your manuscript upon resubmission and further consideration of the manuscript is dependent on the text overlap being addressed in full. Please ensure that your revision is thorough as failure to address the concerns to our satisfaction may result in your submission not being considered further.

Reviewers' comments:

Reviewer's Responses to Questions

**Comments to the Author**

1. Is the manuscript technically sound, and do the data support the conclusions?

Reviewer #1: Yes

Reviewer #2: Partly

2. Has the statistical analysis been performed appropriately and rigorously? 

Reviewer #1: Yes

Reviewer #2: Yes

3. Have the authors made all data underlying the findings in their manuscript fully available?

Reviewer #1: Yes

Reviewer #2: Yes

4. Is the manuscript presented in an intelligible fashion and written in standard English?

Reviewer #1: No

Reviewer #2: Yes

5. Review Comments to the Author

Reviewer #1: Please consider the following suggestions for the manuscript “Carthamus oxyacantha as bio-elicitor and natural defense in maize against charcoal rot disease caused by Macrophomina phaseolina”

Abstract and text:

Line 13: ‘A total of 19 treatments’

Line 17 and onwards: Spell out the word ‘conc.’

Line 16: Rephrase ‘dead inoculum’

SOD, POD, CAT and other acronyms: Do not abbreviate these in the abstract. Instead, write the complete words and their abbreviations the first time they are mentioned in the main text. Use the abbreviations onwards.

Introduction:

Good information about maize and M. phaseolina was provided. Adding some information about C. oxyacantha will be helpful, especially for readers that are not familiar with this weed.

Lines 40 and 45: Not necessary to repeat that M. phaseolina is a seed and soil borne pathogen.

Lines 49-50: Rephrase “climate change is very favorite for M. phaseolina”

Line 66: Spell out the word “1st”

Materials and Methods:

Some sections need additional details for readers to follow how the study was evaluated/conducted.

Lines 69-71: Add specific details/criteria on how Carthamus oxyacantha was identified.

Lines 73-75: How was the pathogen identified as Macrophomina phaseolina?

Lines 75-76: Provide a short summary of the fungal inoculum preparation.

Lines 119-123: What does “G” in equation 2 represent?

Lines 137-142: What equipment/machine was used to measure the pigments?

Discussion:

Indicate author citation for those written as part of the sentence: [36], [39], [40], [43], [46], [12], etc.

Spell out “@”

Proofread the abstract and introduction to make statements clear and unambiguous.

Add line numbers for page 11 and onwards.

Reviewer #2: Manuscript is written decently but lack molecular analysis to support biochemical assay, there is no specific bioelicitor and known marker genes to address the complex network associated with the particular defense mechanism

6. PLOS authors have the option to publish the peer review history of their article (what does this mean?). If published, this will include your full peer review and any attached files.

Reviewer #1: No

Reviewer #2: **Yes: **ALBERT MAIBAM

---

## [Author Response · Author response to Decision Letter 0]

22 Feb 2024

PONE-D-23-34629

Respected Editor (Dr. Abhay K. Pandey) & Reviewers,

The authors are thankful to honorable editor and reviewers for sparing their precious time to review/improve our manuscript “Carthamus oxyacantha as bio-elicitor and natural defense in maize against charcoal rot disease caused by Macrophomina phaseolina”. Keeping in mind the comments of the honorable reviewer 2 (Dr. ALBERT MAIBAM), the authors revised the title to best depict the research work done & presented in this MS as “Allelopathic interactions of Carthamus oxyacantha, Macrophomina phaseolina and maize; Implications for the use of Carthamus oxyacantha as a natural disease management strategy in maize”.

The authors have revised the manuscript in track change mode. 

Request: Please note that the authors have included one more author Anna Andolfi, cosupervisor of PhD thesis of Nazir Aslam. Previously there was a confusion about to include only the name of scholar and the main supervisor in the MS required for the award of PhD degree, now the confusion has been resolved as we need to include the name of cosupervisor also. We realize that PLOS One policy is very strict and there are numerous manuscripts of many authors retracted in the past, so the authors request to accept the inclusion of 3rd author and hope this will not affect the integrity of review process/authorship. Thanks

Reply to the comments are presented below;

Reviewer 1

Abstract and text:

Comment 1. 

Line 13: ‘A total of 19 treatments’

Response: Now it is amended as pointed out.

Comment 2. 

Line 16: Rephrase ‘dead inoculum’

Response: ‘dead inoculum’ is replaced by autoclaved M. phaseolina. It was already discussed as autoclaved M. phaseolina in main body.

Comment 3. 

Line 17 and onwards: Spell out the word ‘conc.’

Response: The authors spelled out the word ‘conc.’ and rephrased it in all the onward text.

Comment 4.

SOD, POD, CAT and other acronyms: Do not abbreviate these in the abstract. Instead, write the complete words and their abbreviations the first time they are mentioned in the main text. Use the abbreviations onwards.

Response: The authors revised and made changes to the manuscript and SOD, POD, CAT and other acronyms fully described at first mention.

Introduction:

Comment 5.

Good information about maize and M. phaseolina was provided. Adding some information about C. oxyacantha will be helpful, especially for readers that are not familiar with this weed.

Response: The authors added information about C. oxyacantha in introduction portion of the manuscript.

Comment 6.

Lines 40 and 45: Not necessary to repeat that M. phaseolina is a seed and soil borne pathogen.

Response: Removed the suggested repeated lines from the manuscript.

Comment 7.

Lines 49-50: Rephrase “climate change is very favorite for M. phaseolina”

Response: Mentioned text have been rephrased.

Comment 8

Line 66: Spell out the word “1st”

Response: Now the word “1st” is replaced by “first”. 

Materials and Methods:

Some sections need additional details for readers to follow how the study was evaluated/conducted.

Comment 9

Lines 69-71: Add specific details/criteria on how Carthamus oxyacantha was identified.

Response: The weed C. oxyacantha was identified based on its vegetative and floral characters and this information is now added in the manuscript.

Comment 11

Lines 73-75: How was the pathogen identified as Macrophomina phaseolina?

Response: The Macrophomina phaseolina was identified by recording its morphological characters such as hyphae shape, colony color, presence of microsclerotia and size of microsclerotia and these specifications were also added in the manuscript.

Comment 12

Lines 75-76: Provide a short summary of the fungal inoculum preparation.

Response: A short summary inoculum preparation has been added in manuscript. 

Comment 13

Lines 119-123: What does “G” in equation 2 represent?

Response: G’ stands for number of grading. This information added in the MS

Comment 14

Lines 137-142: What equipment/machine was used to measure the pigments?

Response: The optical density was recorded at 663, 645, and 440.5 nm wavelength, respectively by using a spectrophotometer (Model UV 3000) and it is also mentioned in the revised manuscript.

Discussion:

Comment 15 Indicate author citation for those written as part of the sentence: [36], [39], [40], [43], [46], [12], etc.

Response: Added author citations for those scientific names written as part of the sentence: [36], [39], [40], [43], [46], [12] and all others at their first mention in the manuscript.

Comment 16

Spell out “@”

Response: Spelled out “@” as “at the rate of” in the manuscript.

Comment 17

Proofread the abstract and introduction to make statements clear and unambiguous.

Response: Proof read and updated the manuscript according to the reviewer’s comments/suggestions.

Comment 18

Add line numbers for page 11 and onwards.

Response: line numbers are added in the revised manuscript.

Reviewer 2

Comment 1

Manuscript is written decently but lack molecular analysis to support biochemical assay, there is no specific bioelicitor and known marker genes to address the complex network associated with the particular defense mechanism.

Response:

The authors are highly obliged for appreciation and encouragement and for providing new insights for related work. The authors modified the title to match the research done as the title was misleading as the reader might think of molecular analysis to support biochemical assays, specific bioelicitor and known marker genes to address the complex network associated with the particular defense mechanism.

Comment 2

Used methods unable to correlate or justify the title.

Response: The title has been modified to depict/justify the research work done.

Comment 3

Try to avoid abbreviation usage in Abstract. Please explain Mp

Response: All abbreviations are now fully described at first mention (abbreviated thereafter) in the abstract as well as in the whole MS. Also pointed out by the reviewer 1.

Comment 4

Estimation of Morphological and biochemical properties generally used as a supplementary data to cross check molecular study. So, addition of molecular analysis (known marker gene) will make complete conclusion.

Response: The confusion arose due to the inappropriate title. Now the title has been revised to represent the actual work done/presented in the MS. By modifying the title comment 2 was also addressed. The authors are highly obliged for useful comment of addition of molecular analysis (known marker gene). The author will consider including such investigations in future studies.

Comment 5

It better to have particular selected active antifungal metabolite

Response:

Antifungal activity of C. oxyacantha has been reported and also the active compounds have been reported as per references 21 & 22. Although, in vitro antifungal activity of C. oxyacantha has been reported, its effectiveness has never been tested against charcoal rot of maize in in-vivo as well as its compatibility for maize were missing also. Therefore, pot experiment was conducted to investigate compatibility, antifungal efficacy of C. oxyacantha, to control charcoal rot of maize caused by M. phaseolina. Moreover, effect of soil amendment with C. oxyacantha, on morphological, physiological, and defense related attributes of maize were also investigated 1st time in this study.

Comments closed:

Regards

Dr. Muhammad Akbar

Associate Professor,

Department of Botany, University of Gujrat, Gujrat, 50700, Punjab, Pakistan.

muhammad.akbar@uog.edu.pk Mobile: +923337645058

---

## [Decision Letter · Decision Letter 1]

25 Mar 2024

PONE-D-23-34629R1Allelopathic interactions of Carthamus oxyacantha, Macrophomina phaseolina and maize; Implications for the use of Carthamus oxyacantha as a natural disease management strategy in maizePLOS ONE

Dear Dr. Akbar,

Thank you for submitting your manuscript to PLOS ONE. After careful consideration, we feel that it has merit but does not fully meet PLOS ONE’s publication criteria as it currently stands. Therefore, we invite you to submit a revised version of the manuscript that addresses the points raised during the review process.

We look forward to receiving your revised manuscript.

Kind regards,

Abhay K. Pandey

Academic Editor

PLOS ONE

Journal Requirements:

Reviewers' comments:

Reviewer's Responses to Questions

**Comments to the Author**

1. If the authors have adequately addressed your comments raised in a previous round of review and you feel that this manuscript is now acceptable for publication, you may indicate that here to bypass the “Comments to the Author” section, enter your conflict of interest statement in the “Confidential to Editor” section, and submit your "Accept" recommendation.

Reviewer #1: All comments have been addressed

Reviewer #2: All comments have been addressed

2. Is the manuscript technically sound, and do the data support the conclusions?

Reviewer #1: Yes

Reviewer #2: No

3. Has the statistical analysis been performed appropriately and rigorously? 

Reviewer #1: Yes

Reviewer #2: No

4. Have the authors made all data underlying the findings in their manuscript fully available?

Reviewer #1: Yes

Reviewer #2: Yes

5. Is the manuscript presented in an intelligible fashion and written in standard English?

Reviewer #1: Yes

Reviewer #2: Yes

6. Review Comments to the Author

Reviewer #1: The authors have addressed my previous comments and made changes to significantly enhance the clarity of the manuscript. Please see minor comments below:

• Title: Use colon instead of semicolon

• Line 28: Change to ‘48.1, 65.3, and 75.0%’ (or 48, 65, and 75%) to make decimal values consistent. Do the same with all values throughout the manuscript.

• Line 98: spell out ‘PDA’

• Line 117: Change to: ‘is given in Table 1.’

• Line 135: ‘between 5 and 6’

• Line 153-155: ‘Morphological parameters were measured after harvesting the plants on 80 DAS. Shoot length, shoot dry weight, and root dry weight of all plants in all treatments were recorded after oven drying at 70 'C until a constant dry weight reading was achieved [31].’

• Line 173: ‘and the following’

• Line 175: ‘the measurements were made between 10:00 AM and 2:00 PM’

• Line 179: ‘The leaf powder was homogenized by adding 3 mL of chilled…’

• Line 180: ‘two centrifuge tubes, the supernatant was centrifuged at…’

• Line 184-185: ‘Crude enzyme solution (50 uL) from each sample…’

• Line 187-189. Rephrase because the sentences read like laboratory protocol.

• Line 196: Rewrite the following phrase because it reads like a laboratory protocol = heat and stir well, add 36.1 uL of 30% H202 after cooling

• Line 203-204: ‘and 1 mL of the reaction solution into the cuvette’

• Line 219: ‘and Mp3 increased progressively with increasing inoculum levels from 60 up to 100% for DI’

• Results: Include the Table or Figure number at the end of first sentence where the result of the Table or Figure is mentioned. This is in addition to the Table or Figure number you included at the end of each paragraph.

• Line 262: ‘Amp2 and Amp3’

• Line 270: ‘A significant decrease of 20…’

• Line 279: ‘and 21%, respectively’

• Discussion: No need to repeatedly write the complete names of previously abbreviated terms.

• Line 345: Write author citation for [45]

Reviewer #2: As you mention, you have refer reference 21 and 22. There they have mentioned clearly the extracted chemical that can further process in detail studies regarding the allelopathic interaction. Why you have used the crude extract as to compare the associated data. It show us that the work undertaken is very preliminary, not so rigorous.

7. PLOS authors have the option to publish the peer review history of their article (what does this mean?). If published, this will include your full peer review and any attached files.

Reviewer #1: No

Reviewer #2: No

---

## [Author Response · Author response to Decision Letter 1]

1 Jun 2024

Response to the Comments 

PONE-D-23-34629R1] 

Respected Editor (Dr. Abhay K. Pandey) & Reviewers,

The authors are thankful to honorable editor and reviewers for sparing their precious time to review/improve our manuscript ““Allelopathic interactions of Carthamus oxyacantha, Macrophomina phaseolina and maize; Implications for the use of Carthamus oxyacantha as a natural disease management strategy in maize”. The authors have revised the manuscript in track change mode. 

Reviewer's Responses to Questions 

Comments to the Author

1. If the authors have adequately addressed your comments raised in a previous round of review and you feel that this manuscript is now acceptable for publication, you may indicate that here to bypass the “Comments to the Author” section, enter your conflict of interest statement in the “Confidential to Editor” section, and submit your "Accept" recommendation.

Reviewer #1: All comments have been addressed

Reviewer #2: All comments have been addressed

2. Is the manuscript technically sound, and do the data support the conclusions?

Reviewer #1: Yes

Reviewer #2: No

Reply: The authors have tried their level best to improve the conclusions section to show more clearly that conclusions have been drawn appropriately based on the data presented.

3. Has the statistical analysis been performed appropriately and rigorously?

Reviewer #1: Yes

Reviewer #2: No

Reply: The authors have tried their level best to choose the appropriate data analysis suitable to show the effects of different treatments on numerous parameters like morphological, physiological, biochemical and disease. These statistical tests included ANOVA followed by Fisher’s LSD test at 5% probability using computer software Minitab 20. (This portion has already included in the MS. The Minitab analysis files are shared as (supporting data). 

The authors also tried to construct the heat map shown below (Shown in word file of response to reviewers) using R programming but the authors realize that this test will not be appropriate to show the effects of different treatments on various parameters and is not suitable for such type of data where multiple treatments with their respective controls are given.

The authors also performed PCA analysis (using OriginPro 2024) of the data that shows the effect of different treatments on various parameters better as compared to heat map, so the authors included the PCA analysis in the revised MS. However, the authors think that graphs with LSD presented better show the effects of different treatments on different parameters.

4. Have the authors made all data underlying the findings in their manuscript fully available?

Reviewer #1: Yes

Reviewer #2: Yes

5. Is the manuscript presented in an intelligible fashion and written in standard English?

Reviewer #1: Yes

Reviewer #2: Yes

6. Review Comments to the Author

Reviewer #1: The authors have addressed my previous comments and made changes to significantly enhance the clarity of the manuscript. Please see minor comments below:

Reply to the comments are presented below;

Reviewer 1

Comment 1. 

Title: Use colon instead of semicolon.

Response: Now it is amended as pointed out and put the colon to replace semicolon.

Comment 2.

Line 28: Change to ‘48.1, 65.3, and 75.0%’ (or 48, 65, and 75%) to make decimal values consistent. Do the same with all values throughout the manuscript.

Response: Now decimal values are consistent throughout the manuscript as per your guidance.

Comment 3.

Line 98: spell out ‘PDA’

Response: PDA changed with potato dextrose agar.

Comment 4.

Line 117: Change to: ‘is given in Table 1.’

Response: “is given in the table 1” have been corrected to ‘is given in Table 1.’ 

Comment 5.

Line 135: ‘between 5 and 6’

Response: “between 5 & 6” have been corrected with ‘between 5 and 6’.

Comment 6.

Line 153-155: ‘Morphological parameters were measured after harvesting the plants on 80 DAS. Shoot length, shoot dry weight, and root dry weight of all plants in all treatments were recorded after oven drying at 70 ºC until a constant dry weight reading was achieved [31].’

Response: The paragraph “Morphological parameters shoot length was measured by measuring tap, shoot dry weight and root dry weight of all plants of all treatments were recorded after oven drying at 70 ºC, till constant dry weight reading, after harvesting on 80 DAS [31]” has been replaced with ‘Morphological parameters were measured after harvesting the plants on 80 DAS. Shoot length, shoot dry weight, and root dry weight of all plants in all treatments were recorded after oven drying at 70 ºC until a constant dry weight reading was achieved [31].’

Comment 7.

Line 173: ‘and the following’

Response: “the” has been added and corrected the sentence as: ‘and the following’ 

Comment 8.

Line 175: ‘the measurements were made between 10:00 AM and 2:00 PM’

Response: “from 10:00 am to 2:00 pm” has been changed with ‘between 10:00 AM and 2:00 PM’

Comment 9.

Line 179: ‘The leaf powder was homogenized by adding 3 mL of chilled…’

Response: The sentence “Homogenized the leaf powder by adding 3 ml of chilled…’ has been corrected and replaced with “The leaf powder was homogenized by adding 3 mL of chilled…’

Comment 10.

Line 180: ‘two centrifuge tubes, the supernatant was centrifuged at…’

Response: ‘two centrifuge tubes, centrifuged at 10,000 x g for 20 min at 4 °C” corrected as ‘two centrifuge tubes, the supernatant was centrifuged at…’

Comment 11.

Line 184-185: ‘Crude enzyme solution (50 uL) from each sample…

Response: the “crude enzyme “has been deleted and now this sentence has been corrected as ‘Crude enzyme solution (50 uL) from each sample…

Comment 12

Line 187-189. Rephrase because the sentences read like laboratory protocol.

Response: Mentioned sentences are now rephrased.

Comment 13.

Line 196: Rewrite the following phrase because it reads like a laboratory protocol = heat and stir well, add 36.1 uL of 30% H202 after cooling.

Response: Mentioned phases are rephrased.

Comment 14

Line 203-204: ‘and 1 mL of the reaction solution into the cuvette’

Response: “were mixed” has been deleted from the sentence and now corrected sentence is “‘and 1 mL of the reaction solution into the cuvette’

Comment 15

Line 219: ‘and Mp3 increased progressively with increasing inoculum levels from 60 up to 100% for DI’

Response: The sentence “and Mp3 were increased progressively with stages from 60 up to 80% for DI “has been replaced with “and Mp3 increased progressively with increasing inoculum levels from 60 up to 80% for DI’. Not 100% as per our disease measurements.

Comment 16

Results: Include the Table or Figure number at the end of first sentence where the result of the Table or Figure is mentioned. This is in addition to the Table or Figure number you included at the end of each paragraph.

Response: Table or Figure number at the end of first sentence included in the revised Manuscript.

Comment 17.

Line 262: ‘Amp2 and Amp3’

Response:. The sign “&“ between Amp2 and Amp3 has been replaced with word “and”.

Comment 18

Line 270: ‘A significant decrease of 20…’

Response: The word “of “has been added before 20 and now updated in the revised manuscript.

Comment 19

Line 279: ‘and 21%, respectively’

Response: “%” has been added in the sentence.

Comment 20

Discussion: No need to repeatedly write the complete names of previously abbreviated terms.

Response: Abbreviations are now fully described at 1st mention in the MS the used only the abbreviations except in table 1 where each species was given with full genus name at first mention, as table needs to be self-explanatory.

Comment 21

Line 345: Write author citation for [45]

Response: Author citation for [45] mentioned in revised manuscript at its first mention in the abstract, then abbreviated onward as per comment 20 above.

Reviewer #2: 

Comment 1

As you mention, you have referred reference 21 and 22. There they have mentioned clearly the extracted chemical that can further process in detail studies regarding the allelopathic interaction. Why you have used the crude extract as to compare the associated data. It shows us that the work undertaken is very preliminary, not so rigorous.

Response:

Reference [21] discusses about antifungal compounds isolated from Cirsium arvense (L). Scop. These authors discussed only the in vitro antifungal activity of compounds isolated from C. arvense. They did not report the in vivo disease suppressing efficacy of C. arvense. (Carthamus oxyacantha was mistakenly mentioned there in the introduction as reference [21] was about C. arvense, while reference [22] was specific to C. oxyacantha, both belonging to family Asteraceae. The authors of reference 22 mentioned the antifungal compound γ-Sitosterol, so reference 22 was retained and old ref [21] was removed and updated latest paper specific to C. oxyacantha.

Only few reports are also available about the presence of antifungal compounds in C. oxyacantha and now we have replaced the reference with reference pertinent to Carthamus oxyacantha. The added reference also report the presence of antifungal compounds in C. oxyacantha just as references [21] in previous version of the MS).

We have not used the crude extract of C. oxyacantha, instead we used the plant biomass as soil amendment.

 In these references researchers mentioned about presence of antifungal compound in C. oxyacantha family Asteraceae, but not antifungal efficacy against charcoal rot causal agent Macrophomina phaseolina, it is first in vivo study in which C. oxyacantha was investigated against charcoal rot of maize. Moreover, the isolated natural compounds can not be tested in vivo because of low quantities purified through chromatographic procedures and these can only be tested in vitro but our present study focused on various other aspects of in vivo studies.

In addition to evaluating the antifungal efficacy at 3 different concentrations, the authors also included treatments to observe the allelopathic effects of not only the autoclaved pathogen but also the weed, C. oxyacantha, not investigated before in detail. Moreover, the authors also confirmed the allelopathic effects in terms of physiochemical attributes, not reported earlier. Therefore, the present study can be viewed as advanced study as it not only extended the studies of earlier researchers but also presented an improved way to investigate the allelopathic interactions. This need was also pointed out in a study reference [23], in a broader sense.

Thanks for your consideration.

Comments closed:

---

## [Editor Report · Decision Letter 2]

1 Jul 2024

Allelopathic interactions of Carthamus oxyacantha, Macrophomina phaseolina and maize: Implications for the use of Carthamus oxyacantha as a natural disease management strategy in maize

PONE-D-23-34629R2

Dear Dr. Akbar,

We’re pleased to inform you that your manuscript has been judged scientifically suitable for publication and will be formally accepted for publication once it meets all outstanding technical requirements.

Kind regards,

Abhay K. Pandey

Academic Editor

PLOS ONE

Additional Editor Comments (optional):

authors have addressed all comments.
---

## [Editor Report · Acceptance letter]

29 Jul 2024

PONE-D-23-34629R2 

PLOS ONE

Dear Dr. Akbar, 

I'm pleased to inform you that your manuscript has been deemed suitable for publication in PLOS ONE. Congratulations! Your manuscript is now being handed over to our production team.

Kind regards, 

on behalf of

Dr. Abhay K. Pandey 

Academic Editor

PLOS ONE